# Methanol-driven esterification of volatile short-chain fatty acids in thermal desorption-based analysis
Philip Kwan Hung Leung [1], Alson Hubert Kwongyiu Wong[1], Yiling Ma[1], Jungmin Jen Yoo[1], María Bajo-Fernández[1,2], Valerio Converso [1], Aaron Parker[1], Patrik Spanel[1,3], George Bushra Hanna[1] & Ilaria Belluomo [1] ✉

Breath volatile organic compound analysis can non-invasively detect diseases, with short-chain fatty acids (SCFAs) identified as key biomarkers. However, SCFA quantification is technically challenging due to chemical instability during thermal desorption (TD) tube analysis. Esterification with methanol may cause methyl ester formation, which impairs diagnostic sensitivity and reproducibility. We hypothesised that methanol-driven esterification of SCFAs is temperature- and time-dependent and can occur under common solvent handling and storage conditions used in TD-based analysis, even without the addition of acid or base catalyst. Here we show that methanol-driven SCFA esterification occurs in the liquid phase but not in the gas phase. Esterification rates increase with higher methanol-to-SCFA ratios and elevated temperatures. Furthermore, prolonged storage at higher temperatures accelerated SCFA esterification, reducing recovery by up to 70% after two months at room temperature and refrigerated conditions. Addressing these artefacts is crucial for ensuring the diagnostic accuracy of SCFA-based breath tests.

Exhaled breath testing, based on the measurement of volatile organic compounds (VOCs), represents a promising non-invasive diagnostic approach[1]. VOCs are small carbon-based molecules with low boiling points and high vapor pressures at ambient temperature. Thousands of different VOCs are reported in exhaled human breath and can be metabolic by-products of mammalian or microbial activity[2,3]. Disease development can alter metabolism and hence the composition of exhaled VOC profiles. Short-chain fatty acids (SCFAs) in particular, are established major metabolic products generated by the gut microbiota and play multiple roles in modulating host-microbe interactions[4]. SCFAs have been reported as biomarkers for gastrointestinal cancers, neurological diseases, lipid oxidation and microbiome dysbiosis[5–9].

Gas chromatography-mass spectrometry (GC-MS) is the gold standard for breath analysis. This typically requires pre-concentration of exhaled breath onto thermal desorption (TD) tubes, which are small stainless-steel cylinders, containing one or more sorbent materials, designed to capture VOCs while allowing other gases to pass through[10]. This enables large-scale, multi-site studies, as the TD tubes can be easily stored and transported, with most VOCs being stable on sorbent for at least 60 days when stored at -80 °C[11,12]. An issue with TD-GC-MS is that it only provides a static snapshot of

VOCs at the time of collection. In contrast, some ambient ionisation mass spectrometry techniques, such as selected ion flow tube-mass spectrometry (SIFT-MS), enable direct, real-time analysis of VOCs without the need for sample preparation[13]. By using different reagent ions, SIFT-MS can continuously monitor a broad range of VOCs within an established, bespoke analytical reference library[14]. Therefore, its main advantage lies in its rapid and dynamic monitoring of VOC compositions[15].

SCFAs are highly volatile and hydrophilic molecules. Their high volatility can be advantageous for gas chromatography but also increases the risk of loss or degradation during suboptimal sample handling[16]. Due to their polarity, GC-MS-based analysis of SCFAs often require pre-column derivatisation to improve chromatographic performance[17]. Due to their hydrophilicity, the presence of water during GC analysis can cause issues such as retention shifts and peak asymmetry. Moreover, studies have reported background noise associated with SCFA measurements, potentially arising from esterification reactions involving methanol, a commonly used and essential solvent in mass spectrometry analytical pipelines[18]. As well as being a good solvent for a wide range of VOCs, its high volatility means it can be purged from sorbent beds without adversely affecting the concentration of other adsorbed VOCs. This property prevents the use of

[1]Division of Surgery, Department of Surgery and Cancer, Faculty of Medicine, Imperial College London, London, UK. [2]Centro de Metabolómica y Bioanálisis (CEMBIO), Facultad de Farmacia, Universidad San Pablo-CEU, CEU Universities, Urbanización Monteprínicipe, Boadilla del Monte, Madrid, Spain. [3]J. Heyrovský Institute of Physical Chemistry, Academy of Sciences of the Czech Republic, Prague, Czech Republic. ✉e-mail: i.belluomo@imperial.ac.uk

alternative solvents without limiting the range of analytes that can be analysed[19].

In the presence of excess methanol, SCFAs undergo Fischer esterification to produce methyl ester and water[20]. These methyl ester products are collectively referred to as fatty acid methyl esters (FAMEs). FAMEs are less polar but more volatile than their parent acids[21]. When TD tubes are spiked with methanol, usually to introduce analytical standards, captured SCFAs from samples may undergo esterification, chemically converting into FAMEs and thus becoming undetectable within the SCFA-specific chromatographic windows[22]. This can result in a substantial underestimation of SCFA concentrations[23]. Several factors are known to influence the production of FAMEs, including the presence of catalysts that can accelerate esterification through acid- or base-catalysed mechanisms[24]. However, few studies have investigated the conditions affecting FAME formation, and SCFAs' recovery losses due to esterification remain poorly characterised in TD-based analysis. Although Fischer esterification is well understood in principle, its extent and kinetics under routine TD tube spiking, storage and handling conditions, at trace analysis relevant levels have not been systematically quantified, and the resulting impact on SCFA recovery and apparent FAME detection in TD-based breath workflows remains underappreciated.

This study aimed to examine the factors that influence the production of FAMEs and hence limit SCFA recovery. We hypothesised that methanol-driven esterification of SCFAs is a temperature- and time-dependent reaction that can occur under common solvent handling and storage conditions used in TD-based VOC analysis, even without the addition of an acid or base catalyst. We show that the methanol-driven esterification of SCFAs can result in a significant depletion of parent acids and the formation of FAMEs. This process is significantly affected by temperature and storage duration.

## Results and discussion
### Methanol-driven SCFA esterification occurs in liquid phase
Direct SIFT-MS monitoring demonstrated that SCFA esterification through reaction with methanol is strictly dependent on the physical state of the reactants. In headspace (gas phase) experiments, 20 µL of SCFA solutions and 20 µL of methanol were placed in separate vials in the microchamber. Upon stabilisation of SCFA signals, no methyl esters were detected over a 60-minute period. SCFA signal intensities remained within ±2% of baseline levels, and no time-dependent increase in methyl ester ions was observed, confirming the absence of esterification under gas-phase conditions (Fig. 1A–F).

In contrast, liquid-liquid phase experiments were performed where 20 µL of methanol and 20 µL of SCFA solution were directly mixed in a single vial. A clear time-dependent increase in methyl ester signals was detected. For example, methyl acetate signal intensity increased by approximately 40% over the first 30 minutes and reached a 2.5-fold increase at 60 minutes relative to baseline. Similar trends were observed for methyl propionate, methyl butyrate, methyl valerate, and methyl hexanoate (Fig. 1G–L).

These findings provide quantitative confirmation that esterification requires direct liquid-phase interaction between methanol and SCFAs and proceeds progressively over time.

### Esterification is dependent on methanol-to-SCFA ratio
Varying the volume ratio of methanol to SCFA further modulated esterification rates. At a 1:1 ratio, maximal production of methyl esters was observed, while excess methanol (9:1) or excess SCFA (1:9) led to a 2–3-fold reduction in methyl ester levels after 60 minutes (Fig. 2). These results indicate that both methanol and acetic acid concentrations influence the esterification reaction, with a greater impact observed from the proportion of acetic acid. These trends are consistent with thermodynamic principles governing Fischer esterification, where optimal concentrations of both reactants facilitate maximal forward reaction rates.

### Temperature- and concentration-dependent esterification of SCFAs
To assess the effect of temperature on the esterification reaction independent of the microchamber conditions, samples were first pre-heated to 50 °C before being placed into the micro-chamber, which was maintained at room temperature. Pre-heating methyl acetate levels were nearly 10-fold higher compared to samples analysed without pre-heating (Fig. 3A, B).

In a complementary experiment, the temperature of the microchamber itself was manually increased by 5 °C every hour to directly assess esterification kinetics during continuous heating. At room temperature, both acetic acid and propionic acid diluted in methanol and water showed a gradual increase in concentration over the first 4 to 5 hours before reaching a plateau. As the temperature increased, distinct spikes in acid concentrations were observed, with more pronounced changes for acetic acid compared to propionic acid. This likely reflects the lower molecular weight, higher volatility, and lower boiling point of acetic acid relative to propionic acid. Moreover, production of FAMEs in methanol increased by nearly 10-fold when the temperature reached 50 °C compared to room temperature (Fig. 3C, D).

Overall, these orthogonal experiments confirm that the esterification of SCFAs by methanol is strongly temperature-dependent, with elevated temperatures accelerating FAME formation.

### SCFA depletion and FAME formation during storage
We then evaluated the long-term stability by assessing the impact of time and temperature on the preservation of SCFAs and the formation of FAMEs. Across all storage conditions, the concentration of all target SCFAs fell below the GC-MS limit of detection after storage. Propionic acid and its corresponding methyl propionate were excluded from further analysis due to co-elution of methyl propionate with a GC column contaminant. SCFAs prepared in water rather than methanol were used as a reaction blank control. No FAMEs were detected at baseline, nor after storage in the water control (Fig. 4A, B).

Time- and temperature-dependent increases in FAME were observed, with a clear dependence on carbon chain length. Storage at room temperature led to a significant increase in all measured FAMEs relative to baseline (time 0), with methyl acetate ($p < 0.01$), methyl butanoate ($p < 0.001$), and methyl valerate ($p < 0.001$) continuing to rise significantly between one and two months. Methyl hexanoate and methyl heptanoate showed smaller increases that plateaued or slightly declined over time. Temperature comparisons further demonstrated that FAME formation was significantly suppressed at lower temperatures, particularly for longer-chain esters ($p < 0.0001$, Fig. 4C).

FAMEs were still detected at −20 °C and −80 °C, indicating that methanol-driven esterification was not eliminated. FAME levels at these temperatures were lower than at 4 °C and room temperature, consistent with kinetic suppression of the reaction (1.5–2-fold lower at −20 °C and 2–4 fold lower at −80 °C compared to 4 °C, depending on carbon chain length). Parent SCFAs were below the limit of quantification under these storage conditions. This loss of parent acids may stem from limitations in TD recovery, the high volatility and polarity of the parent acids, incomplete retention on the sorbent bed (breakthrough) during spiking and dry purging, or reduced desorption and chromatographic performance for the parent acids relative to the corresponding methyl esters. Additionally, calibration of TD tubes for parent SCFAs was not sufficiently robust (nonlinear response and limited reproducibility), which limits confidence in absolute quantification even with internal standard normalisation. Overall, lowering storage temperature reduced FAME formation but did not eliminate it, and parent SCFA quantification remained unreliable under these conditions tested.

Together, these results demonstrate that storage time positively correlates with the production of FAMEs from SCFAs in methanol, and that elevated storage temperatures markedly accelerate esterification.

**Fig. 1 | Real-time detection of headspace and liquid-liquid esterification reactions.**
**A–F** Headspace reactions between methanol and each short-chain fatty acid. **G–L** Liquid-liquid reactions between methanol and each short-chain fatty acid. All experiments were analysed by selected ion flow tube mass spectrometry. *n* = 3. **A, G** Created in BioRender. Leung, P. (2026) https://BioRender. com/zsvw4fi.

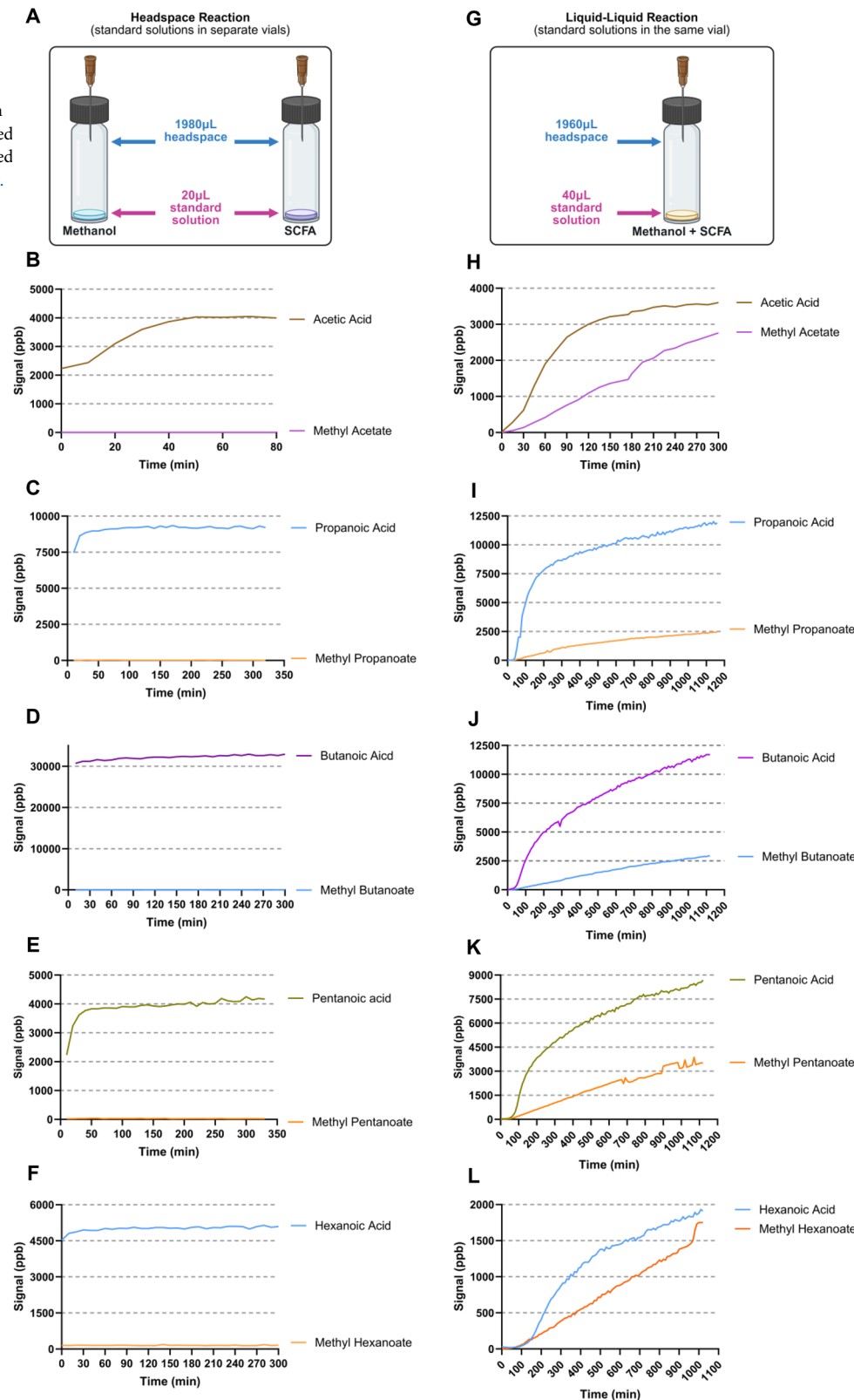

## Conclusion

This study demonstrates that the esterification of SCFAs, driven by methanol, can result in a depletion of parent acids and the formation of FAMEs. An important finding is that this process is significantly affected by temperature and storage duration. While experiments were conducted in liquid-phase systems to isolate and quantify the reaction, it is likely that similar chemical changes occur during the preparation, spiking, or storage of SCFA-containing TD tubes. Recognising and addressing these artefacts is crucial for ensuring the diagnostic accuracy of SCFA-based VOC tests for gastrointestinal cancers and other diseases, and will guide the development of standardised protocols for future clinical applications.

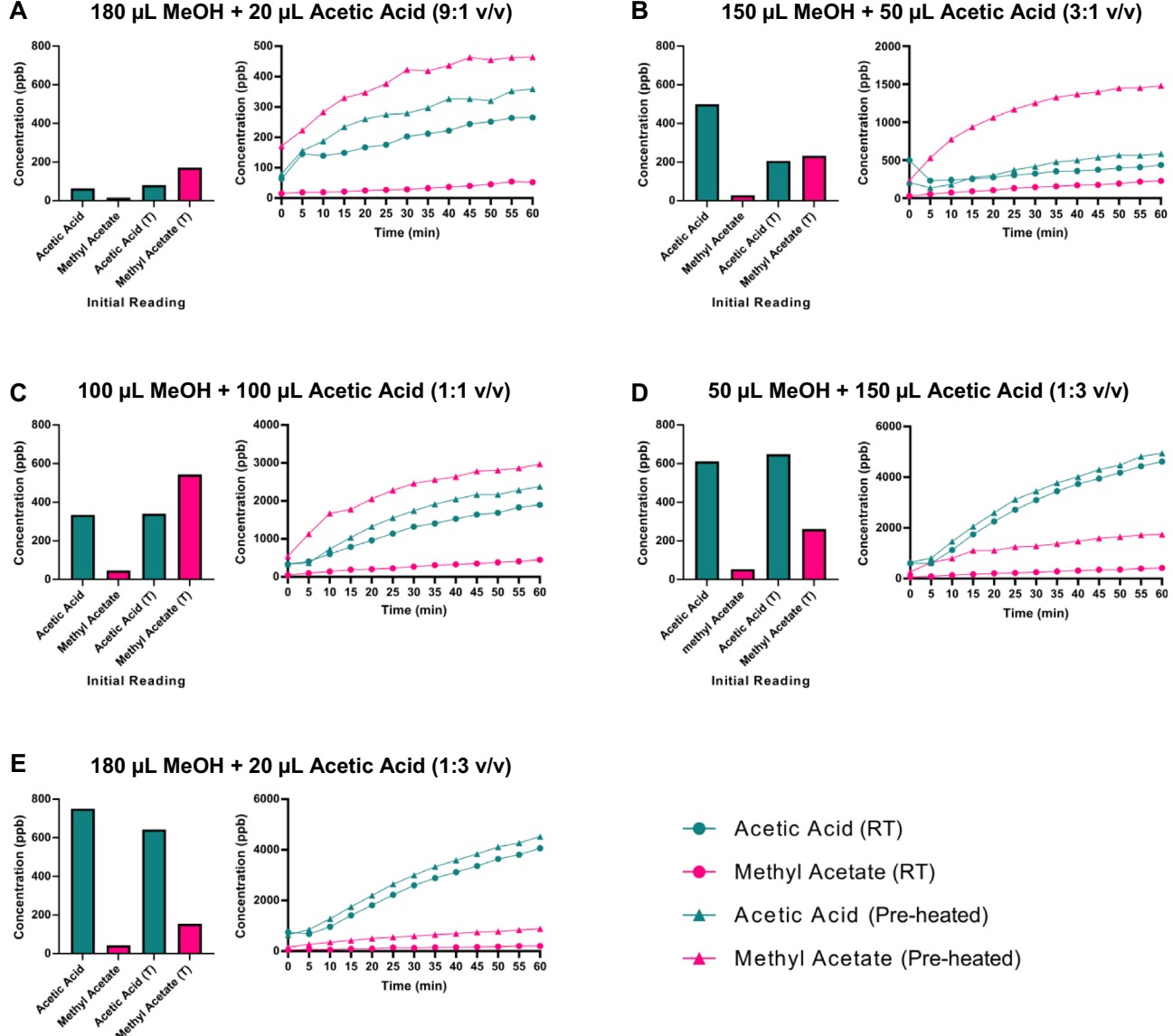

**Fig. 2 | Real-time monitoring of acetic acid and methyl acetate concentrations under varying methanol-to-acetic acid ratios. A–E** Vials were either analysed directly in the microchamber at room temperature (circles), or pre-heated at 50 °C for 1 h before analysis (triangle). Panels represent different methanol-to-acetic acid volume ratios. Bar plots represent the initial baseline reading. All experiments were analysed by selected ion flow tube mass spectrometry. $n = 3$.

## Methods
### Materials and reagents
All SCFA standards, FAME standards and methanol (CAS: 67-56-1) were purchased from Sigma-Aldrich. SCFA standards included acetic acid (CAS: 64-19-7), propionic acid (CAS: 79-09-4), butyric acid (CAS: 107-92-6), valeric acid (CAS: 109-52-4), hexanoic acid (CAS: 142-62-1) and heptanoic acid (CAS: 111-14-8). FAME standards included methyl acetate (CAS: 79-20-9), methyl propionate (CAS: 554-12-1), methyl butyrate (CAS: 623-42-7), methyl valerate (CAS: 624-24-8), methyl hexanoate (CAS: 106-70-7) and methyl heptanoate (CAS: 106-73-0). The internal standard used was toluene-d8 (CAS: 2037-26-5). Type 1 ultrapure water was obtained by reverse osmosis with Milli-Q Direct water purification system (Merck Millipore, Gillingham, UK). An analytical stock solution was prepared in either methanol or water at 100 ppm (100 ng/μL). These analytical stock solutions were used to prepare 9-point calibration curves (0 – 12 ng/μL).

### Selected ion flow tube-mass spectrometry
The Voice200ultra SIFT-MS instrument (Syft Technologies, Christchurch, New Zealand) generates reagent ions by microwave discharge through moist or dry air[25]. Targeted VOCs were quantified based on their known ion-molecule reactions with the selected reagent ions, which formed characteristic product ions (Table 1). These product ions were monitored using a multi-ion monitoring mode. Regular instrument validation was performed according to the manufacturer's protocols. Concentrations of the VOC vapours were estimated from the known reaction time according to the regular SIFT-MS procedure[26].

The SIFT-MS instrument's direct inlet was connected to the microchamber thermal extractor 250 (Marks International, Llantrisant, UK) and constantly supplied with low-flow nitrogen (50 mL/min). Before sampling, the microchamber was conditioned at 200 °C for 30 minutes with a high nitrogen flow (100 mL/min). The temperature was allowed to return to room temperature (22-25 °C) overnight with a continuously supplied nitrogen flow. Direct SIFT-MS was used to monitor the liquid-liquid

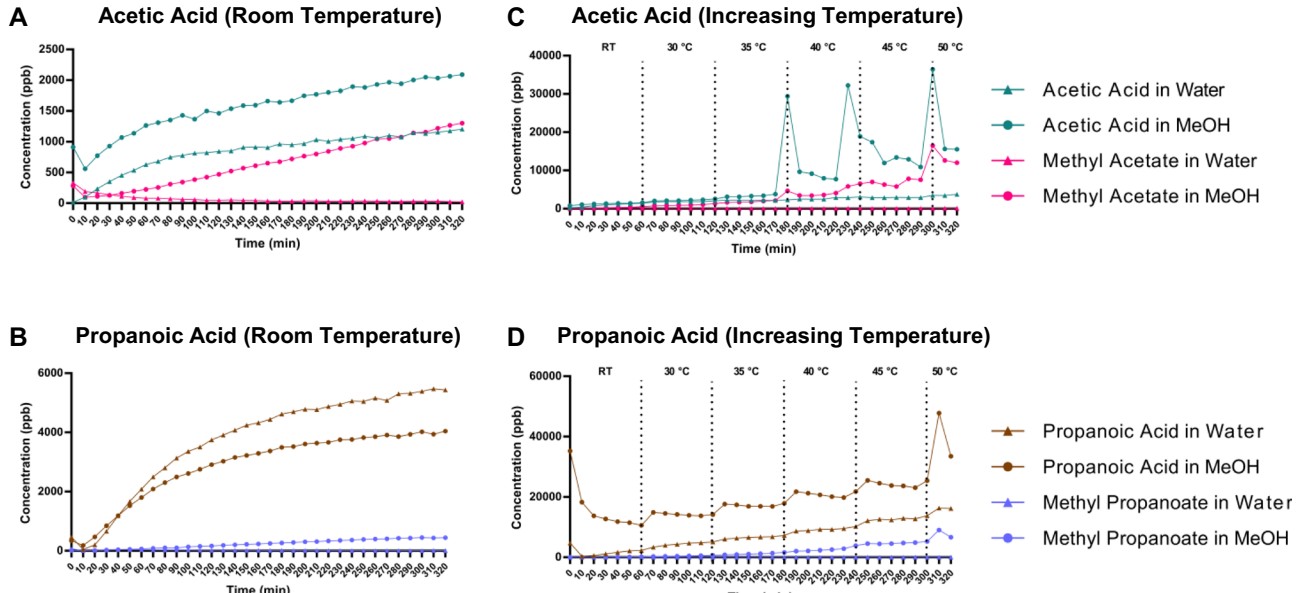

**Fig. 3 | Real-time monitoring of methyl ester production from short-chain fatty acids in methanol or water. A**, **B** Time course of acetic acid and propionic acid when the microchamber was maintained at constant room temperature. **C**, **D** The same reactions with the microchamber temperature manually increased by 5 °C every hour; temperature shifts are indicated by dashed vertical lines. Water was used as a negative reaction control. All experiments were analysed by selected ion flow tube mass spectrometry. $n = 3$.

reaction between methanol and SCFAs through passive diffusion by quantifying the amounts of SCFAs and FAMEs, respectively.

## Microchamber reaction monitoring

Background readings were analysed by direct SIFT-MS prior to monitoring the reactions. In gas-phase reactions, 20 µL of a single SCFA and 20 µL of methanol were added into separate 2 mL vials fitted with needles piercing the septa, allowing passive VOC flow from the liquid inside. The vial containing the SCFA was placed into the microchamber extractor, analysed by direct SIFT-MS. When the signal of the SCFA stabilised for an hour, the vial containing methanol was also placed into the micro-chamber extractor (Fig. 1A, G).

For the liquid-liquid reaction, each vial containing a single SCFA with either methanol or water was placed in the microchamber immediately after preparation and analysed. Water acted as a negative control as SCFAs are not expected to undergo esterification with water. The microchamber temperature increased by 5 °C every hour from room temperature up to 50 °C. Raw data were collected with the batch scanner function every 10 minutes in LabSyft Pro software (version 1.8.1, Syft Technologies, Christchurch, New Zealand). SIFT-MS raw data were analysed using the LabSyft Pro software. The analyte concentrations were exported using the "Data" module.

## Storage stability

Vials containing an analytical mix of SCFA standards prepared in either methanol or water (100 ng/µL) were stored in a temperature-regulated room (21-24 °C), fridge (4 °C), freezer (-20 °C), ultra-low temperature freezer (-80 °C) for up to two months. Quantification of target SCFA and their corresponding FAMEs was performed at baseline (time 0), one month and two months.

## TD tube preparation and spiking

TD tubes were conditioned at 310 °C for 15 min, then background checked using GC-flame ionisation detector (GC-FID). Clean TD tubes were loaded onto a PAL3 Series III Dual Head Sample Handling Robot (SepSolve Analytical). The sample handling robot prepared working concentrations of the calibration mix and

spiked 1 µL of each working calibration standard mix onto TD tubes. Each TD tube was subsequently dry purged for 3 min ($N_2$, 100 mL/min).

## Thermal desorption-gas chromatography-time of flight-mass spectrometry

TD tubes were analysed on two duplicate TD-GC time-of-flight MS (TD-GC-ToF MS) systems, each equipped with mid-polar or polar GC columns respectively. TD tubes were pre-purged with helium (50 mL/min) and then heated at 280 °C for 5 minutes using a TD100-xr (Markes International) to desorb VOCs onto a U-T12ME-2S focusing trap held at 10 °C (Markes International). The focusing trap was then ballistically heated up to 300 °C for 3 minutes with a 5.3:1 inlet:outlet split to release the VOCs into the GC column, with helium carrier gas (1.4 mL/min column flow). For mid-polar measurements, a fused silica capillary column (Rxi-624Sil; 30 m x 0.250 m, 0.25 mm; Restek) was used, while for polar measurements a Stabilwax-DA column (30 m x 0.25 mm, 0.25 mm; Restek) was used. The Agilent 8890 GC oven was kept at 40 °C for 1 minute before raising to 280 °C at a 10 °C/min increase rate with a 10-minute end hold. The GC column was coupled to a BenchTOF2 detector (SepSolve Analytical, Peterborough, UK), with mass range set between 35-500 m/z and a continuous data acquisition rate at 6 Hz.

## GC-MS data analysis

GC-MS raw data were analysed using the ChromSpace software (Sepsolve). Dynamic baseline correction (DBC) was applied to the raw data files at a peak width of 6 seconds. Total peak area quantification was performed using a curve fitting algorithm with 3-point pseudo Gaussian smoothing. All retention times and integration windows were manually adjusted to perform absolute quantification of targeted SCFAs and FAMEs on polar (Table 2) and mid-polar (Table 3) systems respectively. Calibration curves were used for absolute quantification if linear ($R^2 > 0.9$). Absolute analyte amounts (ng) were quantified based on retention time and quant ion (a compound specific m/z fragment from the mass spectrum used for quantitative peak integration) through an automated processing sequence, then normalised against the internal standard toluene-d8.

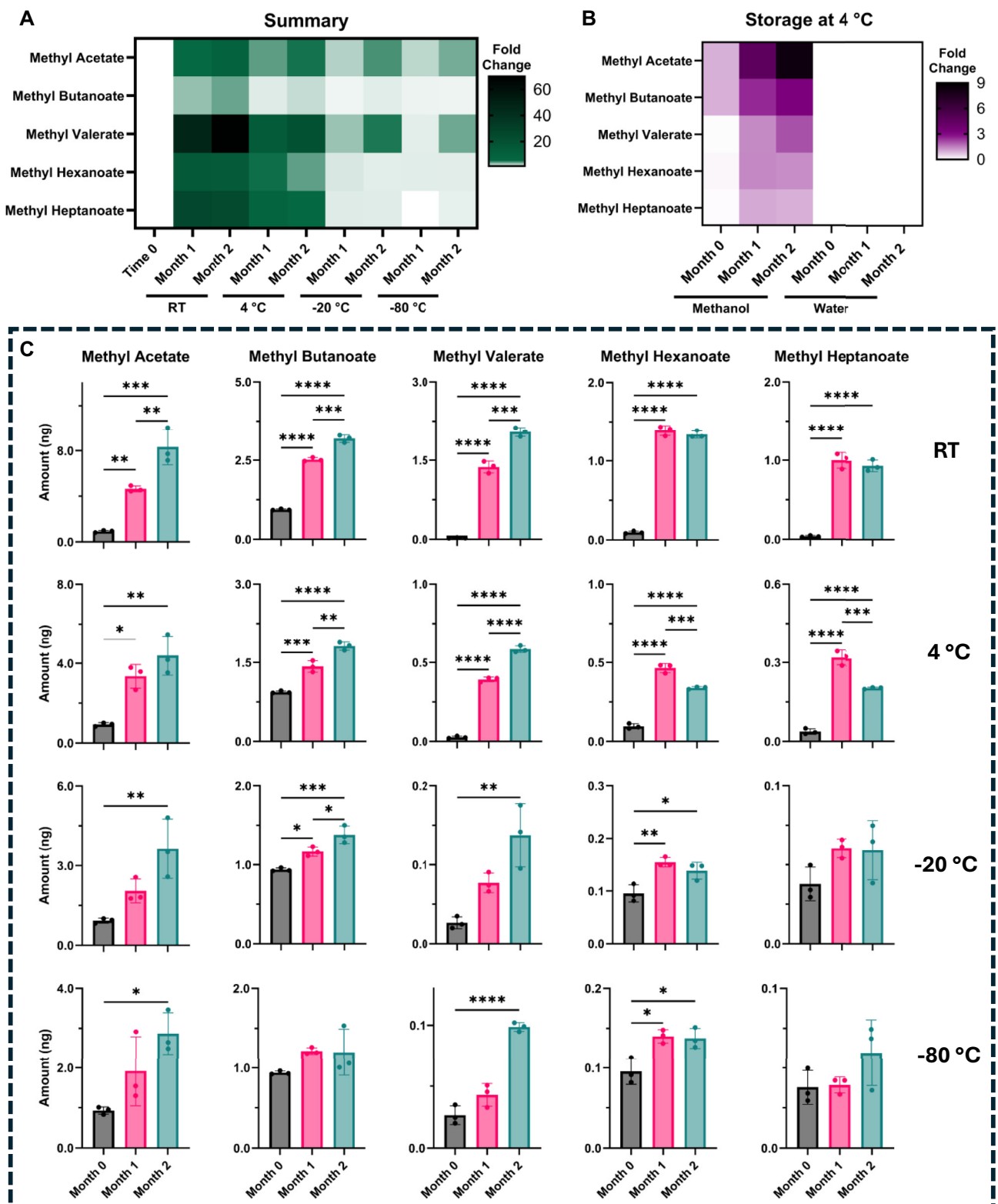

**Fig. 4 | Effects of storage time and temperature on methyl ester formation.**
**A** Summary of methyl esters detected after storage of short-chain fatty acid (SCFA) standards in methanol for 1 and 2 months at room temperature (RT), 4 °C, -20 °C and -80 °C. Data represented as fold change against Month 0. **B** Summary of methyl ester formation in SCFA standards stored at 4 °C in methanol or water. Data represented as fold change against Month 0 of water solution. **C** Absolute quantification (ng) of methyl esters. One-way analysis of variance (ANOVA) followed by Tukey's Honest Significant Difference test was conducted. $p < 0.05$ (*), $< 0.01$ (**), $< 0.001$ (***), $< 0.0001$ (****).

**Table. 1 | Targeted VOCs analysed by selected ion flow tube mass spectrometry**

| Name | Formula | Reagent ions | Product ions | Product ion m/z |
|---|---|---|---|---|
| Methanol | $CH_3OH$ | $H_3O^+$ | $CH_5O^+$ | 33 |
| | | $H_3O^+$ | $CH_3OH_2^+.H_2O$ | 51 |
| Acetic Acid | $CH_3COOH$ | $NO^+$ | $NO^+.CH_3COOH$ | 90 |
| Propionic Acid | $C_3H_6O_2$ | $NO^+$ | $NO^+.C_2H_5COOH$ | 104 |
| Butyric Acid | $C_4H_8O_2$ | $NO^+$ | $NO^+.C_3H_7COOH$ | 118 |
| Valeric Acid | $C_5H_{10}O_2$ | $NO^+$ | $NO^+.C_4H_9COOH$ | 132 |
| Hexanoic Acid | $C_6H_{12}O_2$ | $NO^+$ | $C_6H_{12}O_2.NO^+$ | 146 |
| Methyl Acetate | $C_3H_6O_2$ | $H_3O^+$ | $CH_3COOCH_3.H^+$ | 75 |
| | | $H_3O^+$ | $CH_3COOCH_3.-H^+.H_2O$ | 93 |
| | | $NO^+$ | $NO^+.CH_3COOCH_3$ | 104 |
| Methyl Propionate | $C_4H_8O_2$ | $H_3O^+$ | $C_2H_5COOCH_3.H^+$ | 89 |
| | | $H_3O^+$ | $C_2H_5COOCH_3.-H^+.H_2O$ | 107 |
| | | $NO^+$ | $C_2H_5CO^+$ | 57 |
| | | $NO^+$ | $NO^+.C_2H_5COOOCH_3$ | 118 |
| | | $O_2^+$ | $C_2H_5COOCH_3^+$ | 88 |
| Methyl Butyrate | $C_5H_{10}O_2$ | $H_3O^+$ | $C_3H_7COOCH_3.H^+$ | 103 |
| | | $H_3O^+$ | $C_3H_7COOCH_3.-H^+.H_2O$ | 121 |
| | | $NO^+$ | $C_3H_7CO^+$ | 71 |
| | | $NO^+$ | $NO^+.C_3H_7COOOCH_3$ | 132 |
| | | $O_2^+$ | $C_3H_6O_2^+$ | 74 |
| Methyl Valerate | $C_6H_{12}O_2$ | $H_3O^+$ | $C_6H_{12}O_2.H^+$ | 117 |
| | | $NO^+$ | $C_5H_9O^+$ | 85 |
| | | $NO^+$ | $C_6H_{12}O_2.NO^+$ | 146 |
| | | $O_2^+$ | $C_4H_{10}O^+$ | 74 |
| Methyl Hexanoate | $C_7H_{14}O_2$ | $NO^+$ | $C_6H_{11}O^+$ | 99 |
| | | $O_2^+$ | $C_4H_{10}O^+$ | 74 |

**Table. 2 | Targeted VOC quantification on polar thermal desorption-gas chromatography-time of flight mass spectrometry**

| Name | Retention Time (mins) | Molecular Weight |
|---|---|---|
| Acetic Acid | 9.87 | 60.02 |
| Propionic Acid | 10.95 | 74.04 |
| Butyric Acid | 12.03 | 88.05 |
| Valeric Acid | 13.29 | 102.07 |
| Hexanoic Acid | 14.47 | 116.08 |
| Heptanoic Acid | 15.58 | 130.10 |

All VOCs were integrated using 60 m/z as characteristic Quant Ion

## Statistical analysis

All statistical analyses were performed with GraphPad Prism (v10.4.0, GraphPad software Inc., San Diego, USA). A one-way analysis of variance (ANOVA) followed by Tukey's Honest Significant Difference test was conducted to analyse the effects of storage time on the production of FAMEs. A $p$-value $< 0.05$ was recognised as being statistically significant.

**Table. 3 | Targeted VOC quantification on mid-polar thermal desorption-gas chromatography-time of flight mass spectrometry**

| Name | Retention Time (mins) | Molecular Weight |
|---|---|---|
| Methyl Acetate | 3.20 | 74.04 |
| Methyl Propionate | 4.69 | 88.05 |
| Methyl Butyrate | 6.35 | 102.07 |
| Methyl Valerate | 8.30 | 116.08 |
| Methyl Hexanoate | 10.18 | 130.10 |
| Methyl Heptanoate | 11.96 | 144.12 |

All VOCs were integrated using 60 m/z as characteristic Quant Ion

## Data availability

All data generated or analysed during this study are included in this article, with numerical source data for all figures included under supplementary data 1. Further enquiries can be directed to the corresponding author.

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

## Acknowledgements

This project was supported by Royal Society of Chemistry through the Research Fund grant (R21-0038618851) awarded to IB. PKHL is funded by a Cancer Research UK programme grant awarded to GBH (EDDPGM-May21\100007). IB is funded on an Epilepsy Research Institute UK Emerging Leader Fellowship. MB was supported through an International Research Mobility Grant CEU–Santander. The lab of G.B.H. is supported by programme grants from Cancer Research UK (EDDPGM-May21\100007), Pancreatic Cancer UK (BT2022_Hanna, BT2025_Hanna) and National Institute for Health Research (NIHR207551). Additional laboratory funding originates from the Rosetrees and Stoneygate Trusts, HCA Healthcare UK and infrastructure funding support from the NIHR Imperial Biomedical Research Centre (BRC), and NIHR HealthTech Research Centre in In Vitro Diagnostics (formerly London Medtech and In vitro diagnostic Co-operative).

## Author contributions

Conceptualization & methodology: PKHL, AP, PS, GBH, IB. Investigation: PKHL, AHKW, YM, JJY, VC, MB, IB. Formal analysis: PKHL, AHKW, YM, IB. Project administration: PKHL, IB. Supervision: PS, GBH, IB. Writing – original draft: PKHL. Writing – review & editing: AP, PS, GBH, IB. Funding acquisition & resources: GBH, IB.

## Competing interests

GBH is a founder of a cancer diagnostic company. The remaining authors disclose no conflicts.
