## [Transparent Peer Review file · Communications Chemistry]

Methanol-driven esterification of volatile short-chain fatty acids in thermal desorption-based analysis

Corresponding Author: Dr Ilaria Belluomo

Version 0:

Reviewer comments:

Reviewer #1

(Remarks to the Author)

The paper presents a thorough and detailed examination of esterification on SFCA quantification. The authors have undertaken a very careful, novel and important study, the results of which are of considerable significance to the development of clinical tests that involve exhaled breath volatile analysis. The work will be of considerable interest to the breath research community and to companies developing breath tests, especially for developing non-invasive tests for gastrointestinal diseases. The data analyses have been undertaken using standard and appropriate techniques. The paper has been carefully written, is scientifically correct and is easy to read. I just have a few suggestions for the authors to consider before final publication.

Reviewer #2

(Remarks to the Author)

This paper investigates a critical methodological flaw in the analysis of Short Chain Fatty Acids (SCFAs) within breath analysis workflows. The authors demonstrate that using methanol—a standard solvent in mass spectrometry—causes SCFAs to undergo Fischer esterification, converting them into Fatty Acid Methyl Esters (FAMES). This conversion leads to the underestimation of SCFA biomarkers and the false detection of FAMES, potentially skewing diagnostic results for gastrointestinal and neurological conditions.

I cannot see any novelty in this publication. The core chemical mechanism described—Fischer esterification (Acid + Alcohol \rightarrow Ester + Water)—is a foundational organic chemistry reaction known for over a century. The fact that methanol reacts with carboxylic acids is not new knowledge.

The novelty lies in the quantification of this artifact specifically within the context of Thermal Desorption (TD) tube preparation for breath analysis. While analytical chemists generally know methanol can be reactive, few studies have systematically characterized:

- 1- The rate of this reaction without an acid catalyst in typical storage conditions.
- 2- The specific impact on breath biomarkers (SCFAs) at trace levels.
- 3- The comparison between gas-phase and liquid-phase artifact formation in this specific workflow.

The paper does not discover a new phenomenon but rather provides empirical evidence of a "known unknown" in a specific analytical niche. It validates a suspicion that many researchers may have ignored.

This paper is high impact paper for analytical chemists, metabolomics researchers, and breath scientists. Many researchers use methanol as a solvent to introduce internal standards onto the TD. This paper essentially acts as a "Stop Work" order for that protocol. Also, the research shows that SCFAs that are major biomarkers for cancer, the finding that they disappear or convert to FAMES during storage is critical and can help to explain the inter-lab variability and poor reproducibility in previous studies.

Here are few more questions for authors regarding this research:

- 1- Did author test another solvent beside methanol to find a solution to methanol?
- 2- Authors studied the standards spiked onto tubes. Did they study the effect of humidity (water) that might shift the equilibrium of the esterification?
- 3- Lines 265–271 contain a concerning finding that the authors discuss but do not solve. Even at -80°C , where esterification

was stopped, the parent SCFAs were still undetectable. The authors attribute this to physical volatilization. This dilutes the impact of the paper: even if a researcher eliminates the methanol/esterification issue, the paper suggests the SCFAs might be lost anyway due to handling.

I would recommend this for publication, but likely as a Technical Note or Methodological Communication rather than a full research article, and only after the authors address a specific gap in their data interpretation.

Version 1:

Reviewer comments:

Reviewer #1

(Remarks to the Author)

Thank you for making the minor modifications I suggested. This is a great paper of considerable relevance to the breath volatile research community. Congratulations on this fine work.

Reviewer #2

(Remarks to the Author)

I appreciate the authors' efforts in addressing my previous questions. Despite these clarifications, the broader impact and novelty of this study still appear limited. Because the subject matter will primarily interest a very specific readership, I believe this work would be more appropriately formatted and published as a Short Communication. Thank you.

Methanol driven esterification of volatile short chain fatty acids in thermal desorption-based analysis

Philip Kwan Hung Leung, Alson Hubert Kwongyiu Wong, Yiling Ma, Jungmin Jen Yoo, María Bajo-Fernández, Valerio Converso, Aaron Parker, Patrik Spanel, George Bushra Hanna, Ilaria Belluomo*

*Corresponding Author

Point-by-point response to Reviewers

Reviewer #1 (Remarks to the Author):

The paper presents a thorough and detailed examination of esterification on SFCA quantification. The authors have undertaken a very careful, novel and important study, the results of which are of considerable significance to the development of clinical tests that involve exhaled breath volatile analysis. The work will be of considerable interest to the breath research community and to companies developing breath tests, especially for developing non-invasive tests for gastrointestinal diseases. The data analyses have been undertaken using standard and appropriate techniques. The paper has been carefully written, is scientifically correct and is easy to read.

Author Response: Thank you for the positive comments and appreciation of the work performed.

I just have a few suggestions for the authors to consider before final publication.

Author Response: We have edited the manuscript following your comments, thank you for the suggestions. Here the new version of the sentences incorporating the suggested modifications:

Lines 47-49 *“Here we show that methanol-driven SCFA esterification occurs in the liquid phase but not in the gas phase. Esterification rates increase with higher methanol-to-SCFA ratios and elevated temperatures.”*

Lines 56-57 *“Exhaled breath testing, based on the measurement of volatile organic compounds (VOCs), represents a promising non-invasive diagnostic approach.”*

Lines 66-67 *“Gas chromatography-mass spectrometry (GC-MS) is the gold standard for breath analysis. This typically requires pre-concentration of exhaled breath onto thermal desorption (TD) tubes,”*

Lines 71-72 *“An issue with TD-GC-MS is that it only provides a static snapshot of VOCs at the time of collection.”*

Lines 99-100 *“However, few studies have investigated the conditions affecting FAME formation, and SCFAs’ recovery losses due to esterification remain poorly characterised in TD-based analysis.”*

Lines 106-108 *“We show that the methanol-driven esterification of SCFAs can result in a significant depletion of parent acids and the formation of FAMEs. This process is significantly affected by temperature and storage duration.”*

Lines 128-129 “Concentrations of the VOC vapours were estimated from the known reaction time according to the regular SIFT-MS procedure ²⁶.”

Lines 149-150 “Water acted as a negative control as SCFAs are not expected to undergo esterification with water.”

Lines 175-178 “For mid-polar measurements, a fused silica capillary column (Rxi-624Sil; 30m x 0.25mm, 0.25 mm; Restek) was used, while for polar measurements a Stabilwax-DA column (30 m x 0.25 mm, 0.25 mm; Restek) was used.”

Line 184 “GC-MS raw data were analysed using the ChromSpace software (Sepsolve).”

Line 185-187 “Total peak area quantification was performed using a curve fitting algorithm with 3-point pseudo Gaussian smoothing.”

Line 189 “Calibration curves were used for absolute quantification if linear ($R^2 > 0.9$).”

Lines 198-199 “A p -value < 0.05 was recognised as being statistically significant.”

Lines 69-70: Some mention and reference of the stability of VOCs that are captured on the material in TDs should be provided.

Author Response: We have added additional details on stability with appropriate reference (PMID: 39689424 DOI: 10.1088/1752-7163/ada05c).

Lines 69-71 “This enables large-scale, multi-site studies, as the TD tubes can be easily stored and transported, with most VOCs being stable on sorbent for at least 60 days when stored at $-80\text{ }^{\circ}\text{C}$ ^{11, 12}.”

Line 189-190: “quant ion” This will not be familiar to many readers. Please explain this term.

Author Response: We have added an explanation for this term.

Lines 190-193 “Absolute analyte amounts (ng) were quantified based on retention time and quant ion (a compound specific m/z fragment from the mass spectrum used for quantitative peak integration) through an automated processing sequence, then normalised against the internal standard toluene- d_8 .”

Tables S2 & S3: “Quant Ion column” Is this column necessary given it is the same value throughout? Would it not be sufficient to just mention it in the table caption?

Author Response: We have removed this column and mentioned it in the caption.

Table S2 caption “Table S2. Targeted VOC quantification on polar thermal desorption-gas chromatography-time of flight mass spectrometry. All VOCs were integrated using 60 m/z as characteristic Quant Ion.”

Table S3 caption “Table S3. Targeted VOC quantification on mid-polar thermal desorption-gas chromatography-time of flight mass spectrometry. All VOCs were integrated using 60 m/z as characteristic Quant Ion.”

Reviewer #2 (Remarks to the Author):

This paper investigates a critical methodological flaw in the analysis of Short Chain Fatty Acids (SCFAs) within breath analysis workflows. The authors demonstrate that using methanol—a standard solvent in mass spectrometry—causes SCFAs to undergo Fischer esterification, converting them into Fatty Acid Methyl Esters (FAMEs). This conversion leads to the underestimation of SCFA biomarkers and the false detection of FAMEs, potentially skewing diagnostic results for gastrointestinal and neurological conditions.

I cannot see any novelty in this publication. The core chemical mechanism described—Fischer esterification (Acid + Alcohol \rightarrow Ester + Water)—is a foundational organic chemistry reaction known for over a century. The fact that methanol reacts with carboxylic acids is not new knowledge.

The novelty lies in the quantification of this artifact specifically within the context of Thermal Desorption (TD) tube preparation for breath analysis. While analytical chemists generally know methanol can be reactive, few studies have systematically characterized:

- 1- The rate of this reaction without an acid catalyst in typical storage conditions.
- 2- The specific impact on breath biomarkers (SCFAs) at trace levels.
- 3- The comparison between gas-phase and liquid-phase artifact formation in this specific workflow.

The paper does not discover a new phenomenon but rather provides empirical evidence of a "known unknown" in a specific analytical niche. It validates a suspicion that many researchers may have ignored.

This paper is high impact paper for analytical chemists, metabolomics researchers, and breath scientists. Many researchers use methanol as a solvent to introduce internal standards onto the TD. This paper essentially acts as a "Stop Work" order for that protocol. Also, the research shows that SCFAs that are major biomarkers for cancer, the finding that they disappear or convert to FAMEs during storage is critical and can help to explain the inter-lab variability and poor reproducibility in previous studies.

Author Response: Thank you for your thorough critique. We agree with the above comments and that the importance of the manuscript is to demonstrate such "known unknown" to the wider community and confirm such suspicion in a systematic way, and, for the first time, specifically using TD tubes. To further clarify this point, we added a new sentence:

Line 101 *"Although Fischer esterification is well understood in principle, its extent and kinetics under routine TD tube spiking, storage and handling conditions, at trace analysis relevant levels have not been systematically quantified, and the resulting impact on SCFA recovery and apparent FAME detection in TD-based breath workflows remains underappreciated."*

Here are few more questions for authors regarding this research:

- 1- Did author test another solvent beside methanol to find a solution to methanol?

Author Response: In the paper we did not test any other organic solvent, but we used water as an alternative. However, we recognise that many standards will not be soluble in water hence it is not a good option. Moreover, it will affect the TD-GC-TOF-MS analysis as the water needs to be purged off prior to analysis. This purging process has the risk of removing highly volatile compounds from the TD before analysis which is not ideal.

2- Authors studied the standards spiked onto tubes. Did they study the effect of humidity (water) that might shift the equilibrium of the esterification?

Author Response: We used water as control (Figures 3, 4) as SCFAs are not expected to undergo esterification with water. As supported by these figures, there was no esterification of SCFA in water, thus this would also represent the effect of humidity.

3- Lines 265–271 contain a concerning finding that the authors discuss but do not solve. Even at -80°C, where esterification was stopped, the parent SCFAs were still undetectable. The authors attribute this to physical volatilization. This dilutes the impact of the paper: even if a researcher eliminates the methanol/esterification issue, the paper suggests the SCFAs might be lost anyway due to handling.

Author Response: Thank you for pointing this out. We have clarified this in the revised manuscript to highlight that the esterification was not stopped at -80 °C as there were still FAMEs detected. However, the extent of esterification was highly reduced as seen by the lower levels compared to Room temperature and 4 °C. Parent SCFAs were below the limit of quantification, as the quantification performed building a calibration curve in TD tubes with internal standard was not reliable and linear. Future work would seek to understand what strategies could potentially be done to overcome this issue. We have reworded the sentence for better clarity:

Lines 268-273 "FAMEs were still detected at -20 °C and -80 °C, indicating that methanol-driven esterification was not eliminated. FAME levels at these temperatures were lower than at 4 °C and room temperature, consistent with kinetic suppression of the reaction (1.5-2 fold lower at -20 °C and 2–4 fold lower at -80 °C compared to 4 °C, depending on carbon chain length). Parent SCFAs were below the limit of quantification under these storage conditions. This loss of parent acids may stem from limitations in TD recovery, the high volatility and polarity of the parent acids, incomplete retention on the sorbent bed (breakthrough) during spiking and dry purging, or reduced desorption and chromatographic performance for the parent acids relative to the corresponding methyl esters. Additionally, calibration of TD tubes for parent SCFAs was not sufficiently robust (nonlinear response and limited reproducibility), which limits confidence in absolute quantification even with internal standard normalisation. Overall, lowering storage temperature reduced FAME formation but did not eliminate it, and parent SCFA quantification remained unreliable under these conditions tested."